# Standardized Nursing Terminologies and Electronic Health Records: A Secondary Analysis of a Systematic Review

**DOI:** 10.3390/healthcare13161952

**Published:** 2025-08-09

**Authors:** Luca Bertocchi, Cristina Petrucci, Vittorio Masotta, Alessia Marcotullio, Dorothy Jones, Loreto Lancia, Angelo Dante

**Affiliations:** 1Hematology Unit, Trieste University Hospital (ASUGI), Piazza dell’Ospitale 1, 34125 Trieste, Italy; luca.bertocchi@asugi.sanita.fvg.it; 2The Marjory Gordon Program for Clinical Reasoning and Knowledge Development, William F. Connell School of Nursing, Boston College, Chestnut Hill, MA 02467, USA; jones@bc.edu; 3Department of Health, Life and Environment Sciences, University of L’Aquila Edificio Rita Levi Montalcini, Via G. Petrini, 67100 L’Aquila, Italy; cristina.petrucci@univaq.it (C.P.); vittorio.masotta1@univaq.it (V.M.); alessia.marcotullio@hotmail.it (A.M.); loreto.lancia@univaq.it (L.L.)

**Keywords:** standardized nursing terminologies, nursing assessment, electronic health records, outcomes, Gordon’s functional health patterns model

## Abstract

**Background/Objectives:** Standardized nursing terminologies (SNTs) have been associated with improved patient and organizational outcomes. This secondary analysis aims to examine how structured nursing assessment data and documentation are integrated into electronic health records (EHRs) in studies that report on the impact of American Nurses Association–recognized SNTs. **Methods:** Data were extracted from all 53 primary studies included in a previously published systematic review. The original literature search was conducted in PubMed, Scopus, CINAHL, and OpenGrey. Extracted data focused on nursing assessment tools, use of EHRs, inter-rater reliability, and methodological characteristics. **Results:** Gordon’s Eleven Functional Health Patterns was the most frequently used nursing assessment framework, often in combination with NANDA-I diagnoses. However, details regarding assessment tools and their application in EHRs were inconsistently reported. Only about one-third of the studies explicitly indicated the use of EHRs, though an upward trend in their use has been observed over the last decade. Inter-rater reliability was reported in a limited number of studies, with considerable variation. An overall increasing trend in the use of nursing assessment data in electronic health records was observed over the past decade. **Conclusions:** The integration of SNTs with structured assessment frameworks into EHRs is increasing but remains inconsistently reported. Standardized documentation practices could strengthen nursing visibility, support quality improvement, and enhance outcome measurement in both clinical and research contexts.

## 1. Introduction

Healthcare systems can transform practice and lead to improved patient outcomes by delivering high-quality, evidence-based care guided by a person-centered approach [1]. Nurses, who represent the largest segment of the healthcare workforce [2], play a vital role in providing individualized, person-centered care and in driving improvements in health outcomes and system transformation [1].

A person-centered care approach is often central to the mission of nursing and other healthcare providers and can be effectively integrated into an organization’s strategic goals. However, without reliable nursing data, it becomes challenging to evaluate the unique and collaborative contributions of nurses and their impact on patient outcomes. This challenge can be addressed by incorporating standardized nursing terminologies (SNTs) [3] into healthcare documentation systems, which help highlight nurses’ contributions to person-centered care. Literature reviews have demonstrated the beneficial effect of using SNTs to describe patient and organizational outcomes [4,5,6,7,8,9,10]. By enabling a clearer connection between nursing interventions and outcomes, SNTs may also support the identification of gaps or omissions in care delivery, such as missed nursing care [11].

Using a common SNT provides descriptive data to describe clinical phenomena of concern to nurses within the nursing process framework [12]. This nursing process is a dynamic and iterative six-step clinical reasoning process that includes nursing assessment, diagnosis, outcomes identification, planning, implementation, and evaluation. Use of this process allows nurses to integrate critical thinking to identify and then manage patients’ responses to a human health condition [3].

Nursing assessment represents a foundational part of quality nursing care [13]. The collection of clinical data and information related to health promotes the identification of the patient’s responses to disease [3]. The nursing assessment allows nurses to collect pertinent data about health, resulting in accurate nursing diagnoses that can guide evidence-driven interventions and evaluate data-driven, evidence-based outcomes. A comprehensive nursing assessment makes possible accurate nursing diagnoses and guides the selection of interventions that lead to beneficial outcomes [13,14,15]. The twelve SNTs (Table 1) recognized by the American Nurses Association [16] support the diagnosis, outcome, and intervention stages of the nursing process [13,17,18], although a standardized framework for the assessment phase remains lacking.

In recent decades, the use of electronic health records (EHRs) has become an important priority for documenting healthcare [7]. A meaningful use of health information technology represents an opportunity to communicate and transform practice [1,19]. To effectively use health information technology, adopting standardized nursing data in EHRs, especially for assessments, can help nurses to document patient care and predict and measure patient outcomes of care [18,19].

Previous reviews have described the impact and effectiveness of using SNTs on patients and organizational outcomes [4,5,6,7,8,9,10,16]; however, these discussions have not focused on how the use of standardized nursing assessment criteria is integrated into EHRs. This is a critical communication gap, as the structured documentation of nursing assessment data within the EHRs supports consistency, interoperability, and quality data to support clinical decision-making. Existing reviews tend to generalize findings around outcomes or terminologies without addressing the specific tools used for assessment or the degree of EHR integration. This study addresses this gap by providing a focused synthesis of the literature on how SNTs are used to document nursing assessments and how this information is incorporated into EHR systems.

This study aims to provide insights into the integration of structured nursing assessment data into EHRs by synthesizing the existing literature. It focuses on studies that examine the use of SNTs to document patient information and organizational outcomes, while also analyzing their methodological features and emerging trends. To guide the analysis, the study is structured around the following research questions: (1) Which nursing assessment measures or evaluation tools and criteria have been used to identify nursing diagnoses? (2) Are the data collected integrated into EHRs? (3) What are the methodological characteristics of the studies, particularly regarding inter-rater reliability, study focus, and type of analysis? (4) What trends can be observed in terms of publication outcomes in studies that use SNTs and integrate them into EHRs?

For clarity, in this study we use the following definitions: ‘structured nursing assessment’ refers to the systematic and standardized collection of patient data to support accurate nursing diagnoses and care planning [13,20]; ‘documentation’ denotes the formal recording of nursing assessments, diagnoses, interventions, and outcomes, typically within EHRs, to ensure clear communication and data integrity [21]; and ‘methodological characteristics’ encompass key features of the included studies, such as design, data collection methods, reliability measures (e.g., inter-rater reliability), and analysis type, which influence the validity and applicability of findings [4,5].

## 2. Materials and Methods

This study is based on a secondary analysis of a previously published systematic review [22]. The original review was conducted and reported in accordance with the Preferred Reporting Items for Systematic Reviews and Meta-Analyses (PRISMA) guidelines (see Appendix A) [23,24]. Literature search strategies and data abstraction procedures were established to guide the selection and synthesis of studies, as detailed in the original publication and its supplementary materials [9].

### 2.1. Literature Search Strategies, Inclusion Criteria, and Data Abstraction

The original literature search was conducted in four electronic databases, i.e., PubMed, Scopus, CINAHL, and OpenGrey, using predefined strategies that combined terms related to standardized nursing terminologies (SNTs) and patient or organizational outcomes. Full search strings and methodological details are available in the original systematic review and its supplementary materials [9].

For this secondary analysis, studies included in the original systematic review were further considered based on their relevance to structured nursing assessment and integration into EHRs.

All studies had been originally selected according to the following eligibility criteria: (1) reference to at least one of the 12 ANA-recognized SNTs used to describe patient or organizational outcomes; (2) full-text availability; (3) publication in English or Italian; (4) use of experimental, quasi-experimental, or observational study designs; (5) inclusion of adult or pediatric patients; and (6) analysis of associations between SNTs and outcomes. For the purposes of this secondary analysis, the full texts of all included studies were re-examined to identify whether a structured nursing assessment model (e.g., Gordon’s Functional Health Patterns, Henderson’s Need Theory) was employed and whether any mention of integration into or use of EHRs was reported. This allowed for the classification of studies according to the presence of structured assessment models and EHR-related documentation.

Data were extracted by two researchers independently through a previously tested electronic spreadsheet (Microsoft Excel^®^ Version 2506 Build 16.0.18925.20076). Any disagreement about data extraction was discussed and resolved through discussion with a third author. Extracted data included (a) nursing assessment criteria used, (b) the modality and type of nursing documentation used (paper or electronic format), (c) inter-rater reliability of the SNTs, and (d) publication characteristics such as year to identify emerging trends. Inter-rater reliability is defined as the nurses’ agreement in making nursing diagnoses, nursing outcomes, or nursing interventions [25]. Data were then synthesized in both narrative and tabular forms. Ethical approval and informed consent were not required for this type of study design.

In the original systematic review [9], the quality appraisal of included studies was conducted using validated tools appropriate to study design: the Revised Cochrane Risk of Bias tool (RoB-2) for randomized trials, ROBINS-I for non-randomized studies, and the Joanna Briggs Institute checklist for cross-sectional studies. Two independent reviewers assessed the risk of bias, with disagreements resolved through discussion with a third reviewer until a consensus was reached. While studies with a high risk of bias were excluded from the meta-analysis, all studies were included in the narrative synthesis and, accordingly, were retained in this secondary analysis. Risk of bias results were summarized using ROBVIS 0.3.0^®^ software.

### 2.2. Data Analysis and Synthesis

The data set consisted of quantitative data, which were analyzed using descriptive statistical methods aligned with the specific research questions. Data analysis was conducted using the Statistical Package for Social Science (SPSS), Version 28.0 software (IBM Corp., Armonk, NY, USA). Results were synthesized narratively, in tabular format, and through graphical representations to illustrate key findings related to nursing assessment criteria, documentation formats, inter-rater reliability, and publication trends across the included studies.

## 3. Results

All 53 studies included in the original systematic review have also been analyzed in this secondary study. The main characteristics of these studies are reported in the original publication [9], while the present analysis focuses on the use of reported nursing assessment tools, the integration of nursing assessment and documentation into EHRs, methodological aspects, and emerging trends, as detailed below.

### 3.1. Nursing Assessment Tools and Criteria Supporting Nursing Diagnoses

Among the 53 studies analyzed, several incorporated SNTs within structured nursing assessment frameworks, most notably the Functional Health Pattern Assessment framework, into electronic health records (EHRs).

The analysis showed that Gordon’s Eleven Functional Health Patterns (FHPs) tools were used the most frequently (*n* = 10), especially in studies over the last decade, during which time the integration with EHRs became more widespread [26,27,28,29,30,31,32,33,34,35]. The FHP framework was predominantly used in studies along with NANDA-I nursing diagnoses, with one exception: O’Brien-Pallas et al. (2001) [35], who employed the Omaha System. Onori (2013) [36] used Virginia Henderson’s Need Theory, while the remaining studies (*n* = 42) did not clearly specify an assessment tool or mentioned it only briefly (Table 2). In terms of nursing assessment criteria, five studies supported nursing diagnoses using both ‘signs and symptoms’ and ‘etiology,’ in accordance with the problem-etiology-signs and symptoms (PES) format [28,29,34,37,38]. In four studies [26,39,40,41], the NANDA-I nursing diagnoses were supported by defining characteristics. Erci (2012) [42] and Ning (2021) [43] reported the identification of ‘signs and symptoms.’ Lastly, Paans and colleagues (2010) [21] did not specify the use of ‘signs and symptoms’ or ‘etiology’ but employed the D-Catch instrument, which is based on the PES format, to assess the accuracy of nursing documentation (Table 2) [21].

### 3.2. Integration of Nursing Assessment and Documentation into Electronic Health Records

Approximately one-third of the studies (*n* = 17) utilized EHRs (Table 2). Analysis of these studies revealed an overall increasing trend over time in the number of studies incorporating nursing data into EHRs (Figure 1).

In more than half of the studies (*n* = 31), the type of nursing documentation used to support data reported was not specified. Three studies reported the use of barcode readers [36,67,71], while two studies indicated that data were collected manually [41,48]. Notably, among the 10 studies using Gordon’s FHPs, five are included in the 17 studies that utilized EHRs. This indicates that nearly one-third of the EHR-based studies incorporated Gordon’s FHPs as the nursing assessment framework (Table 2). Additionally, three Italian studies reported the use of the ‘nursing assessment form,’ a clinical decision support system embedded within a nursing information system known as the ‘professional assessment instrument’ [29,33,34].

### 3.3. Methodological Characteristics of Studies Using Standardized Nursing Terminologies

Inter-rater reliability was assessed in 16 of the included studies (Table 2). The inter-rater reliability of nursing diagnoses showed considerable variability, with values ranging from 47% [32] to nearly 98% [53] (Table 2). Marek (1996) [58] established a cut-off of at least 80% inter-rater reliability before nurses could use the Omaha System in documentation. Castellan and colleagues (2016) [28] were the only authors to measure, in addition to nursing diagnoses, inter-rater reliability for nursing outcomes and nursing interventions. Using the Nursing Minimum Data Set, Smith (1994) [41] observed an inter-rater reliability of 97.9%. Finally, inter-rater reliability of the Nursing Outcomes Classification ranged from 70 to 89% [28] to 81% [57], with at least 88.9% in all outcome measures [59] and over 90% [68]. These values reflect the level of agreement among nurses in assigning standardized nursing diagnoses, outcomes, or interventions using SNTs. The inter-rater reliability results provide important evidence that SNT terms are applied consistently, supporting the validity of data collection guided by these terminologies.

With regard to the focus of the studies, the majority (*n* = 25) examined a possible or causal relationship between nursing care using the SNTs and the patient and organizational outcomes, 23 explored the predictive power of nursing diagnoses, and five investigated an association between SNTs and outcomes (Table 2). It should be noted that among these 25 studies, only 17 used EHR data, while the remaining 8 relied on other documentation methods such as paper-based or hybrid systems. This difference in data sources may introduce some heterogeneity, which was considered when interpreting the findings.

In terms of statistical analyses, more than half of the studies (*n* = 27) used multivariate analysis (such as regressions, logistic analysis of variance, analysis of covariance, multiple analysis of variance, and generalized estimating equation) as the primary method, while 26 employed bivariate analysis (such as *t*-tests and chi-squared tests) in addition to other demographic statistics (Table 2). These statistical tests were applied primarily to evaluate the relationships between SNT-based nursing documentation and patient or organizational outcomes rather than to directly test the SNT variables themselves. In most of the studies, SNT variables such as nursing diagnoses, interventions, or outcomes were used as independent variables in statistical models, while patient or organizational outcomes were treated as dependent variables. While statistical analysis is not strictly required to verify ontology use in documentation, the studies included in this review primarily aimed to assess the relationships between SNT-guided documentation and clinical or organizational outcomes, thereby justifying the application of such analyses.

An increasing trend in the number of studies was observed over the decades, particularly in those investigating cause–effect relationships (Figure 2).

Power analysis (sample size calculation) was undertaken in 22 studies to determine the appropriate number of participants required to ensure sufficient statistical power, and eight of these studies used specific power analysis software: WinPepi 10.5 or 11.43 [40,45,47], PASS 11 [54,75,76], and G-Power 3 [42,68]. In the remaining studies, the sample size was calculated using a formula (Table 2).

### 3.4. Trends in Outcomes Research Studies Using Standardized Nursing Terminologies

The studies spanned a 36-year period, from 1985 to 2021, showing an overall increasing trend over time, especially in the last decade (Figure 3).

## 4. Discussion

The findings from this investigation provide a comprehensive synthesis of the literature on the use of SNTs in relation to (a) nursing assessment tools and criteria used to support nursing diagnoses; (b) the integration of structured nursing data into EHRs; (c) the methodological characteristics of the studies, including inter-rater reliability and data analysis approaches; and (d) key patterns observed across publications over time.

### 4.1. Nursing Assessment Tools and Criteria Supporting Nursing Diagnoses

Several structured nursing assessment tools are available and could be integrated into clinical practice to enhance data collection and generate more accurate, nurse-centric data to guide patient care [20]. Among these, the most frequently used tool identified in this review was Gordon’s Eleven FHPs assessment framework [79], most often applied in conjunction with the NANDA-I nursing diagnosis taxonomy.

According to a recent integrative review [13], although a few comprehensive assessment tools have been developed based on Gordon’s FHPs, one with particularly strong evidence of validity is the Functional Health Pattern Assessment Screening Tool (FHPAST) [80]. Future outcomes research could benefit from adopting either Gordon’s framework or this screening tool to further explore the impact of nursing care on patient and organizational outcomes.

Another framework featured in this review was Virginia Henderson’s Need Theory, used by Onori (2013) [36]. It is noteworthy that Gordon reviewed Henderson’s work during the development of her own framework, shifting the focus of assessment from a needs-based approach to a problem-oriented perspective. For example, rather than framing a patient’s concern as ‘I need medication for my pain,’ the approach emphasizes identifying the underlying problem, such as the nursing diagnosis of ‘ineffective pain self-management.’ While Henderson’s Need Theory has historical significance in nursing theory and education, it is less commonly integrated with SNTs or embedded into EHRs, which may limit its usability in contemporary data-driven practice environments.

Gordon’s FHPs framework has been widely adopted to guide data collection and facilitate the identification and organization of nursing diagnoses [12]. It has also supported investigations into individuals responses to illness, for instance, anxiety during the COVID-19 pandemic [81], and enhanced nursing students’ application of the nursing process [82]. Furthermore, evidence suggests that using Gordon’s framework in clinical practice is associated with improved patient outcomes, including better control of depressive symptoms and hopelessness, improved quality of life, and reduced hospital admissions, morbidity, and mortality [13,14,15].

Given these favorable outcomes, a nursing-focused assessment framework such as Gordon’s FHPs may help address gaps in patient data collection [18,79] and provide validated evidence of the assessment data to support the development and naming of the nursing diagnoses along with guiding evidence to increase the accuracy of interventions identified to improve patient quality care outcomes. Integrating this framework with ANA-recognized SNTs could improve the organization of patient information and ultimately contribute to enhanced patient and organizational outcomes. Additionally, the use of structured nursing assessment tools and recognized SNTs may expand the generation of big nursing databases in EHRs.

Within this review, some studies followed the PES format [79], explicitly specifying the related factors (etiology) and defining characteristics (signs and symptoms) for each nursing diagnosis. However, most studies did not report the assessment criteria or the specific nursing assessment instruments used. Notably, the study by Paans et al. (2016) [64] was the only one to employ an instrument specifically designed to evaluate the accuracy of nursing documentation, i.e., the D-Catch tool.

The use of SNTs enhances the accuracy of nursing documentation, which is critical for both reflecting the care provided and ensuring data reliability for measuring and improving outcomes such as patient satisfaction, pressure injury prevention, and nutritional care [83].

Although Gordon’s FHPs framework dominated the studies included in this review as the primary structured nursing assessment tool, the Omaha System (recognized as an SNT) also demonstrated significant potential in supporting comprehensive nursing care and improving outcomes [9]. The Omaha System in the original broader review revealed strong support for its effectiveness, including studies employing rigorous designs such as randomized controlled trials. Unlike Gordon’s FHPs, which are specifically designed to guide the nursing assessment phase, the Omaha System encompasses all components of the nursing process within a fully integrated structure. This makes it difficult to isolate the assessment component alone but highlights its strength as a holistic, practice-oriented system. Its design enhances interoperability and facilitates documentation in EHR environments, making it highly suitable for generating structured, nurse-sensitive big data. Notably, O’Brien-Pallas et al. (2001) [35] applied both the Omaha System and Gordon’s FHPs in tandem, suggesting these models can be complementary, with FHPs guiding comprehensive data collection and the Omaha System supporting standardized documentation and outcome measurement across the continuum of care.

### 4.2. Integration of Nursing Assessment and Documentation into Electronic Health Records

Despite advancements in technology, our investigation revealed that only a small proportion of studies reported using EHRs. Specifically, approximately 30% of the analyzed studies indicated the use of electronic support systems. Although the percentage differs, the findings align with the general direction reported by De Groot et al. (2020), who noted that only half of the respondents in a web-based survey utilized SNTs within EHRs [84].

Nonetheless, our analysis also identified a gradual upward trend in the adoption of SNTs within EHRs. Over time, an increasing number of studies have used EHRs to assess the impact of SNTs on outcomes, corroborating previous literature [5]. Among the 17 studies using electronic health records, five employed Gordon’s FHPs, making it the most frequently declared assessment framework within this subgroup. This finding underscores the continued relevance of Gordon’s model in guiding structured nursing assessment, even in digital documentation environments [85]. These findings reflect a longstanding transition from paper-based and narrative nursing documentation toward the integration of SNTs in EHRs, a process that has unfolded over several decades [5,86].

The global expansion of EHRs for documenting care represents a valuable opportunity to generate evidence describing nursing care and its impact on patient outcomes. The growing use of EHRs and nursing information systems, combined with the availability of larger nursing datasets, enables the retrieval, aggregation, and analysis of nursing data. This facilitates the meaningful application of SNTs to quantify nursing’s contribution to care, link the occurence of nursing diagnoses and their holistic focus on the patient to improved patient outcomes, and promote the use of data as a way to describe nursing complexity [87].

To realize this potential, sustained efforts are required to enable the meaningful use of EHRs through the implementation of nursing clinical information systems that support standardized and interoperable clinical nursing data [19]. Emerging technologies, including artificial intelligence and machine learning, offer powerful tools for predictive analytics of nursing data and the formulation of nursing diagnoses [88,89,90,91]. Studies utilizing nursing information systems have been able to collect more data and perform analysis on larger datasets. This approach may yield stronger evidence of the relationship between SNTs and outcomes, as current evidence on the efficacy of nursing record systems remains limited and inconclusive.

An important consideration when integrating SNTs into EHRs is the significant overlap among the various nursing terminologies. Although these ontologies often have distinct structures or focus areas, such as diagnoses, interventions, or outcomes, they frequently describe similar clinical concepts using different terms. This semantic overlap means that treating them as entirely separate systems may be misleading. Instead, emphasis should be placed on their interoperability and mapping, which are critical for ensuring consistent, accurate nursing documentation and facilitating meaningful use of nursing data within electronic systems [92,93].

### 4.3. Methodological Characteristics of Studies Using Standardized Nursing Terminologies

Inter-rater reliability refers to the degree of agreement between two or more observers (nurses) operating independently to assign values on a scale, such as formulating diagnoses or applying diagnostic rules [25]. Poor diagnostic reliability in nurses’ clinical judgment (i.e., nursing diagnoses) could pose a serious problem, as inaccurate diagnoses may lead to inappropriate interventions or misinterpretation of related outcomes [34,94]. Inter-rater reliability for SNTs was reported in a minority of the studies, with values ranging from moderate to high. Specifically, nursing diagnoses showed reliability from 47% [32] to 98% [53], and the Nursing Outcomes Classification from 70% to over 90%. These results provide additional insights beyond a previous review [4], particularly concerning outcomes and interventions. The variability in inter-rater reliability may impact or threaten the robustness and generalizability of findings, as differences in training, context, or clinical experience can lead to inconsistent use of SNTs across settings. As a consequence, the results of studies should be interpreted carefully in the absence of clarity about the diagnostic criteria or the agreement among nurses on nursing diagnosis identification [4,5]. Strengthening inter-rater calibration and promoting standardized documentation practices could help improve the methodological rigor of future studies and support more consistent implementation of SNTs in clinical practice. Since inter-rater reliability was reported in only a minority of the included studies, caution is warranted when interpreting the overall methodological quality, as the lack of systematic reporting may indicate heterogeneity in study rigor and underscores the need for more transparent and standardized documentation in future research. In addition, it should be noted that among the 25 studies examining cause–effect relationships, only 17 used EHR data, while the remaining 8 relied on alternative documentation sources. This variation in data sources may affect data comparability and should be considered when interpreting the strength of the findings.

The majority of studies reviewed focued on examining the cause–effect relationship between SNTs use and outcomes, as well as the predictive power of nursing diagnoses. Numerous studies explored this predictive capability using different regression models. The most frequently utilized statistical method was bivariate analysis, although several studies also employed multiple regressions to examine relationships between the dependent variable and two or more independent variables, identifying factors contributing to specific outcomes. These findings align with previous reviews [4,5], although a slight increase in the use of bivariate analysis was observed compared to Sanson and colleagues (2017), who reported multiple regressions as the most common analytical model. This shift could be attributed to the growing number of randomized controlled trials conducted in recent years. In randomized controlled trials, multiple regressions are often unnecessary, as confounding variables are already adjusted for through the control group and the randomization process.

Finally, power analysis was conducted in fewer than half of the studies, indicating that this methodological element warrants greater attention in future research.

### 4.4. Trends in Outcomes Research Studies Using Standardized Nursing Terminologies

There has been a noticeable increase in the number of articles published about the impact of SNTs on patients and organizational outcomes. This positive trend aligns with a previous review by Tastan et al. (2014) [16] and highlights the growing research interest in the use of SNTs to improve patient care and enhance the visibility of nursing’s contributions to healthcare delivery.

### 4.5. Implications for Practice, Policy Development, and Future Research

The findings of this study have significant implications for nursing clinical practice, health policy, and future research. First, this investigation offers deeper insight into studies that have investigated the clinical effectiveness of SNTs. These findings can support healthcare policymakers and nursing leaders in recognizing the value of standardized nursing assessments and the structured data needed to demonstrate nursing’s contributions to patient and organizational outcomes. For example, in clinical settings, nurse managers could implement standardized templates within EHRs based on structured nursing assessment frameworks to guide initial patient assessments. This approach would promote consistency across units and improve data quality. In terms of education, incorporating mandatory continuing training on the use of SNTs may help nurses develop accurate nursing diagnoses and appropriately link them to outcomes and interventions. These sessions could be reinforced through case-based simulations embedded in electronic systems [94].

Moreover, operationalizing ANA-recognized terminologies in clinical practice could include embedding taxonomies such as NANDA-I, NIC, and NOC directly into electronic charting templates. This integration would enable real-time decision support, allowing nurses to select standardized diagnoses, interventions, and expected outcomes from dropdown menus during documentation. Such systems can enhance documentation accuracy and interoperability while also supporting audit trails, quality monitoring, and outcome-based reimbursement models [7,95].

Second, the results from this study could inform healthcare policymakers to require the documentation of standardized nursing assessment data within EHRs. Institutionalizing such practices would enhance the visibility of nursing’s impact and enable more comprehensive, data-driven clinical decision-making.

Finally, this study offers a foundation for future research on the use of SNTs. Future research is needed to evaluate the impact of SNTs on outcomes, particularly using structured nursing assessment tools integrated into EHRs. In addition, future primary studies and systematic reviews aim to strengthen the evidence base supporting the role of nursing assessment in improving outcomes and further explore best practices for incorporating assessment data documentation into EHRs.

### 4.6. Strengths and Limitations

A major strength of this study lies in its ability to address a gap in the literature by providing detailed, updated insights on the use of SNTs. These findings may support policymakers and nursing leaders, helping to highlight and support the visibility of nursing contributions to outcomes.

Although the review employed rigorous methods, relevant studies may have been excluded due to the selection of only four databases and the restriction to English and Italian language publications, potentially omitting valuable research in other languages or sources. Furthermore, as this is a secondary analysis, the results are limited by the design and scope of the original systematic review. Specifically, some relevant studies might not have been captured, and the metadata available for analysis may have been incomplete. These aspects should be considered when interpreting the results.

It is also important to note that all studies included in this review utilize at least one SNT recognized by the ANA, according to our inclusion criteria. Consequently, this review does not provide insights into nursing documentation practices that do not employ SNTs. This limitation restricts our ability to compare documentation patterns and outcomes between SNT and non-SNT approaches. Future research is warranted to explore nursing documentation without SNT use, which would provide a broader perspective on nursing documentation practices and their impact on patient and organizational outcomes.

Despite this limitation, the findings offer meaningful insights into nursing assessment tools, EHR integration, and methodological approaches in outcomes research using SNTs.

## 5. Conclusions

This study provided an overview of how SNTs are integrated with structured nursing assessment and EHRs, while also exploring their methodological features and trends in published literature. Gordon’s Eleven FHPs emerged as the most frequently reported assessment framework, although the tools and systems used were often insufficiently described. Only a limited number of studies explicitly reported the use of EHRs and inter-rater reliability measures, suggesting a need for more transparent and standardized documentation practices. Over the decades, a growing number of studies have addressed these topics, indicating increasing interest in the integration of nursing assessment, SNTs, and digital health systems. Promoting the visibility of nursing contributions through consistent use of SNTs and structured assessment within EHRs may enhance the quality of care and support outcome evaluation.

## Figures and Tables

**Figure 1 healthcare-13-01952-f001:**
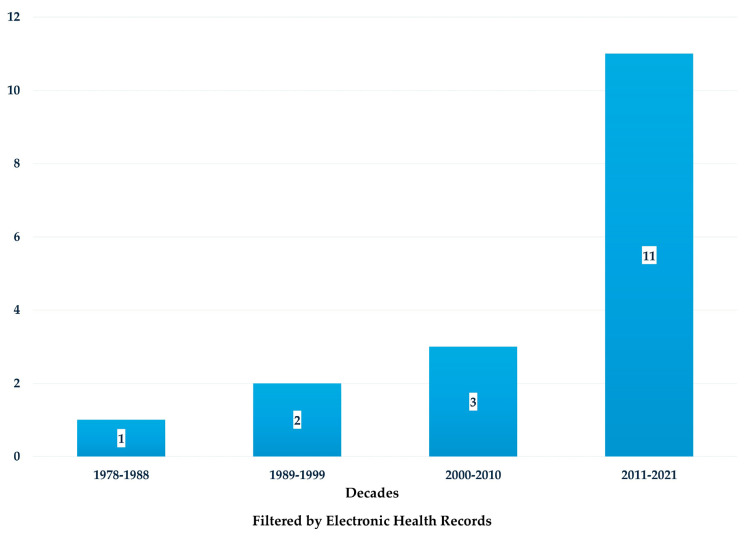
Number of articles for the decade about the impact of SNTs on patient and organizational outcomes for electronic health records.

**Figure 2 healthcare-13-01952-f002:**
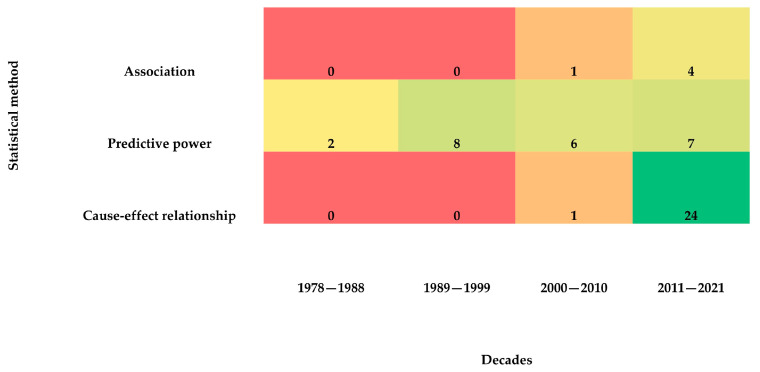
Number of articles per decade by the type of study focus that was used in studies reporting the impact of American Nurses Association–recognized SNTs. Cell color represents frequency intensity, ranging from red (low frequency) to green (high frequency), with yellow and orange indicating intermediate values.

**Figure 3 healthcare-13-01952-f003:**
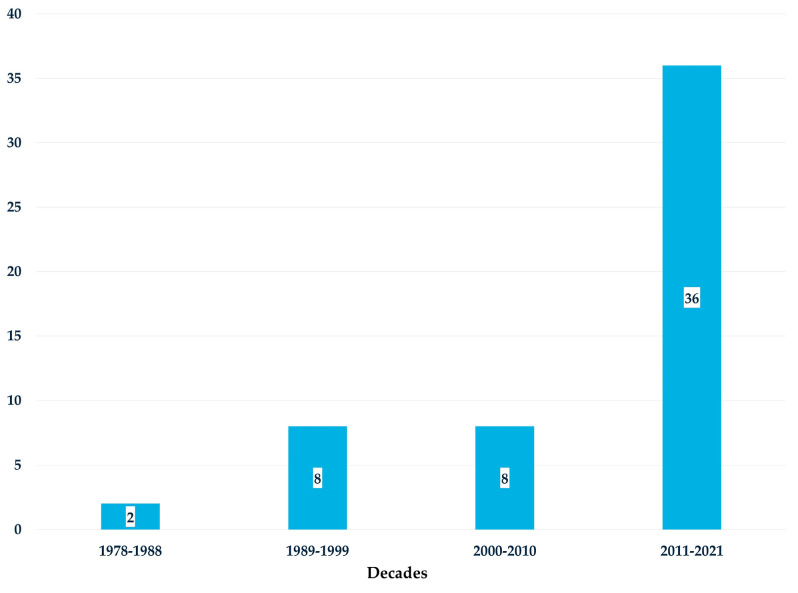
Number of articles for each decade about the impact of SNTs on patient and organizational outcomes.

**Table 1 healthcare-13-01952-t001:** Standardized nursing terminologies recognized by the American Nurses Association.

Standardized Nursing Terminologies	Stage of Nursing Process
NANDA-International (NANDA-I)	Diagnosis
Nursing Intervention Classification (NIC)	Intervention
Nursing Outcome Classification (NOC)	Outcome
Clinical Care Classification (CCC)	Diagnosis, Intervention, Outcome
Omaha System (OS)	Diagnosis, Intervention, Outcome
Perioperative Nursing Data Set (PNDS)	Diagnosis, Intervention, Outcome
International Classification for Nursing Practice (ICNP)	Diagnosis, Intervention, Outcome
Systemized Nomenclature of Medicine Clinical Terms (SNOMED-CT)	Diagnosis, Intervention, Outcome
Logical Observation Identifiers Names and Codes (LOIC)	Diagnosis
Nursing Minimum Data Sets (NMDS)	-
Nursing Management Minimum Data Sets (NMMDS)	-
Alternative Billing Concepts Codes (ABC Codes)	-

**Table 2 healthcare-13-01952-t002:** Summary of findings about nursing assessment, type of health records in nursing documentation, and methodological criteria of the included studies.

First Author, Year	SNT	Nursing Assessment		HR Type	Methodological Criteria			
		Nursing Assessment Tool	PES Format		Inter-Rater Reliability	Study Focus	Main Statistical Analyses (Tests)	Power Analysis
Akkuş, 2012 [44]	NANDA-I	✓ Not specified	X	X	X	Cause–effect	Bivariate (chi-squared test, Wilcoxon ranked test, Mann–Whitney U-test)	✓
Amad Pastor, 2017 [26]	NNN	✓ Gordon’s functional health patterns	✓ Signs and symptoms (DC)	X	X	Cause–effect	Bivariate (Wilcoxon signed-rank test non-parametric)	X
Azzolin, 2013 [37]	NNN linkages	X	✓ Etiology (RF) + signs and symptoms (DC)	X	X	Cause–effect	Multivariate (generalized estimating equations)	✓
Azzolin, 2015 [45]	NNN	X	X	X	X	Cause–effect	Multivariate (generalized estimating equations, Pearson correlation coefficient)	✓
Cardenas-Valladolid, 2012 [27]	NANDA-I, NIC	✓ Gordon’s functional health patterns	X	X	X	Cause–effect	Multivariate (analysis of covariance, logistic regression)	X
Castellan, 2016 [28]	NANDA-I	✓ Gordon’s functional health patterns	✓ Etiology (RF) + signs and symptoms (DC)	X	✓ 78–95% (NANDA); 70–89% (NOC); 66–90% (NIC)	Predictive	Multivariate (multiple linear and logistic regression)	✓
Company-Sancho, 2017 [46]	NANDA-I	X	X	✓ Electronic	X	Predictive	Multivariate (multiple linear regressions)	X
D’Agostino, 2017 [34]	NANDA-I	✓ Gordon’s functional health patterns	✓ Etiology (RF) +signs and symptoms (DC)	✓ Electronic	X	Association	Bivariate (Spearman’s correlation)	X
D’Agostino, 2019 [29]	NANDA-I	✓ Gordon’s functional health patterns	✓ Etiology (RF) + signs and symptoms (DC)	✓ Electronic	X	Predictive	Multivariate (ordinary least squares linear regression models)	X
da Silva, 2015 [47]	NANDA-I, NOC	X	X	✓ Electronic	X	Cause–effect	Bivariate (Student’s *t*-test for paired samples)	✓
da Silva, 2019 [38]	NNN	X	✓ Etiology (RF) + signs and symptoms (DC)	X	X	Cause–effect	Bivariate (Pearson’s chi-squared test, Fisher’s exact test)	X
Erci, 2012 [42]	Omaha System	✓ Not specified	✓ Signs and symptoms	X	X	Cause–effect	Bivariate (paired *t*-test, Pearson correlation)	✓
Gencbas, 2018 [48]	NNN linkages	X	X	✓ Manually	X	Cause–effect	Bivariate (chi-squared test, Mann–Whitney U-test, Student’s *t*-test, Wilcoxon test)	✓
Halloran, 1985 [39]	NANDA-I	X	✓ Signs and symptoms (DC)	X	✓ 91%	Predictive	Multivariate (stepwise regression analysis)	X
Halloran, 1987 [30]	NANDA-I	✓ Gordon’s functional health patterns	X	✓ Electronic	✓ >90%	Predictive	Multivariate (stepwise linear regression analysis)	X
Hays, 1992 [49]	Omaha System	X	X	X	X	Predictive	Multivariate (regression)	✓
Helberg, 1994 [50]	Omaha System	X	X	X	81%	Predictive	Multivariate (hierarchical regression analysis)	X
Juve-Udina, 2017 [51]	NANDA-I	X	X	X	X	Association	Bivariate (chi-squared test, Student’s *t*-test, or Mann–Whitney U test)	✓
Laguna-Parras, 2013 [52]	NNN	X	X	X	X	Cause–effect	Multivariate (multivariate analysis of variance)	✓
Lee, 2002 [53]	NANDA-I	X	X	X	✓ Close to 98%	Predictive	Multivariate (multiple stepwise regression)	X
Lemos, 2020 [40]	NNN	X	✓ Signs and symptoms (DC)	X	X	Cause–effect	Bivariate (Pearson’s correlation)	✓
Liu, 2020 [54]	Omaha System	X	X	X	X	Cause–effect	Multivariate (repeated-measures ANOVA, chi-squared test, or independent *t*-test)	✓
Liu, 2021 [55]	Omaha System	X	X	X	X	Cause–effect	Bivariate (independent *t*-tests, chi-squared tests)	X
Lunney, 2004 [56]	NNN	X	X	X	X	Cause–effect	Bivariate (two-group *t*-tests)	X
Luz-Rodríguez-Acelas, 2020 [57]	NANDA-I, NOC	X	X	X	✓ 81% (nursing outcomes NOC)	Association	Bivariate (Student’s *t*-test)	✓
Marek, 1996 [58]	Omaha System	X	X	X	✓ Cut-off of at least 80%	Predictive	Multivariate (hierarchical regression analysis)	X
Mello, 2016 [59]	NNN	X	X	X	✓	Cause–effect	Multivariate (generalized estimating equation)	✓
Morales-Asencio, 2009 [60]	NANDA-I, NOC	X	X	✓ Electronic	X	Predictive	Multivariate (linear and logistic regression models)	X
Naughton, 1999 [61]	NANDA-I	X	X	X	X	Predictive	Multivariate (logistic regression)	X
Ning, 2021 [43]	Omaha System	✓ Not specified	✓ Signs and symptoms	X	X	Cause–effect	Multivariate (repeated-measures ANOVA)	X
O’Brien-Pallas, 1997 [31]	NANDA-I	✓ Gordon’s functional health patterns	X	X	X	Predictive	Multivariate (stepwise regression analysis)	X
O’Brien-Pallas, 2001 [35]	Omaha System	✓ Gordon’s functional health patterns	X	X	X	Predictive	Multivariate (hierarchical regression analysis)	X
O’Brien-Pallas, 2002 [62]	Omaha System	X	X	✓ Electronic	X	Predictive	Multivariate (hierarchical linear and linear regression)	X
O’Brien-Pallas, 2010 [63]	NANDA-I	X	X	X	X	Predictive	Multivariate (hierarchical regression linear modeling)	X
Onori, 2013 [36]	NANDA-I	✓ Virginia Henderson’s Need Theory	X	✓ Barcode reader	X	Predictive	Multivariate (multiple linear regression, factorial ANOVA)	X
Paans, 2016 [64]	NANDA-I	X	X (but used of the D-Catch)	✓ Electronic	X	Predictive	Multivariate (logistic regression; Poisson regression)	X
Park, 2019 [65]	Omaha System	X	X	✓ Electronic	X	Association	Multivariate (analysis of covariance)	X
Pérez Rivas, 2016 [66]	NNN	X	X	✓ Electronic	X	Cause–effect	Bivariate (chi-square, Student’s *t*-test)	X
Rosenthal, 1992 [32]	NANDA-I	✓ Gordon’s functional health patterns	X	✓ Electronic	✓ 47% (fair)	Predictive	Multivariate (multiple linear regression, factorial ANOVA) and ROC	X
Rosenthal, 1995 [67]	NANDA-I	X	X	✓ Barcode reader	X	Predictive	Multivariate (multiple linear regression, factorial ANOVA) and ROC	X
Sampaio, 2018 [68]	ICNP/NNN	X	X	X	✓ 99–100% (NOC: Anxiety level); 92–98% (NOC: Anxiety self-control)	Cause–effect	Multivariate (multiple linear regression)	✓
Sanson, 2019 [33]	NANDA-I	✓ Gordon’s functional health patterns	X	✓ Electronic	X	Predictive	Multivariate (logistic regression) and ROC	X
Schein, 2005 [69]	NIC	X	X	X	X	Association	Multivariate (standard multiple linear and logistic regression models, simple regression)	X
Smith, 1994 [41]	NANDA-I, NMDS	✓ Not specified	✓ Signs and symptoms (DC)	✓ Manually	✓ 97.9% (NMDS)	Predictive	Multivariate (multiple linear regression)	✓
Wei, 2019 [70]	Omaha System	X	X	X	X	Cause–effect	Bivariate (*t*-test)	X
Welton, 2005 [71]	NANDA-I	X	X	✓ Barcode reader	X	Predictive	Multivariate (logistic regression model; c statistic)	X
Wong, 2015 [72]	Omaha System	X	X	X	X	Cause–effect	Multivariate (repeated measures ANOVA with intention-to-treat strategy)	✓
Xiao, 2019 [73]	Omaha System	X	X	X	X	Cause–effect	Multivariate (ANOVA, independent-sample *t*-test)	X
Zeffiro, 2020 [74]	NANDA-I	X	X	X	X	Predictive	Multivariate (backward stepwise logistic regression model)	X
Zhang, 2017 [75]	Omaha System	X	X	X	X	Cause–effect	Bivariate (chi-squared tests, paired or independent *t*-test)	✓
Zhang, 2018 [76]	Omaha System	X	X	X	X	Cause–effect	Bivariate (chi-squared tests, paired or independent *t*-tests)	✓
Zhao, 2020 [77]	Omaha System	X	X	X	X	Cause–effect	Bivariate (*t*-test)	✓
Zhuang, 2021 [78]	Omaha System	✓ Not specified	X	✓ Electronic	X	Cause–effect	Bivariate (*t*-tests, chi-squared tests)	X

Abbreviations. ANOVA, Analysis of Variance; DC, Defining Characteristics; HR, Health Records; NANDA-I, NANDA-International; NOC, Nursing Outcome Classification; NIC, Nursing Intervention Classification; NNN, NANDA-I-NIC-NOC; NMDS, Nursing Minimum Data Set; PES format, Problem, Etiology—Signs and Symptoms; RF, Related Factors; SNT, Standardized Nursing Terminology.

## Data Availability

No new data were created.

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
