# Peer review of "Standardized Nursing Terminologies and Electronic Health Records: A Secondary Analysis of a Systematic Review"

_healthcare, 2025, doi:10.3390/healthcare13161952_

Round 1

Reviewer 1 Report

Comments and Suggestions for Authors

 1. Your manuscript references the importance of SNTs and integration with EHRs. It would benefit from a more precise articulation of the specific knowledge gap this secondary analysis addresses. The authors should explain why existing reviews are insufficient and how this study adds value.

 2. The authors should provide more detailed descriptions of how data were extracted from the primary studies. For example, mention whether a quality appraisal was performed on the original studies and how discrepancies during data extraction were resolved.

 3. Although the authors relied on a previously published systematic review, the inclusion criteria could be recapped more explicitly for transparency. I suggest you emphasize how the studies were filtered for relevance to structured nursing assessment and EHR integration.

 4. The authors mentioned variability in inter-rater reliability but didn’t critically evaluate its implications. I suggest you discuss how this variability may affect the robustness or generalizability of findings across different healthcare settings.

 5. Although your manuscript includes some figures, it would benefit from additional visuals such as:
 5.1 A matrix or flow diagram showing studies using specific SNTs and EHRs.
 5.2 A heatmap or timeline visualizing methodological trends (e.g., types of statistical analysis) across decades.

 6. The discussion touches on practice and policy implications but remains general. I suggest you consider adding concrete examples of how findings could be operationalized in clinical settings (e.g., through policy mandates or training programs).

 7. Because this study is a secondary analysis, I suggest you acknowledge more explicitly any limitations imposed by the design of the original systematic review, such as potential bias in the original study selection or limitations in available metadata.

 8. Terms like “structured nursing assessment,” “documentation,” and “methodological characteristics” are used variably. To avoid reader confusion, the author should define these upfront and use them consistently throughout the manuscript.

 9. The authors stated that Gordon’s FHPs dominate the findings. I suggest you include a comparative discussion with other frameworks (e.g., Omaha System, Virginia Henderson) regarding coverage, usability in EHRs, and impact on outcomes would be valuable.

 10. The authors cite many foundational studies. Some sections (particularly on AI and EHR integration) would benefit from citations to recent literature (2022–2025) on real-world applications of SNTs in predictive analytics, big data nursing, or machine learning.

Reviewer 2 Report

Comments and Suggestions for Authors

Review: EHR and Standardized Nursing terminology to give better systematic review results.

Focus on documentation practices in regard to SNT. 

EHR mentioned in title but only 17/53 of the studies have EHR.

A previous systematic review of 53 papers was revisited with a focus on examining nursing related SNT and other data documentation. 

There are 12 different SNT ontologies that are reported on here. They do not all apply to the same stages of the nursing process.

Question: Do these ontologies overlap? They are describing the same underlying medical conditions. How do they overlap? Is it meaningful to discuss them as distinct when they are correlated? 

This study aims to provide insights into the integration of structured nursing assessment data into electronic health records (EHRs) by synthesizing the existing literature. 

The research questions of interest are: (1) Which nursing assessment tools and criteria have been used to identify nursing diagnoses? (2) Are the data collected integrated into EHRs? (3) What are the methodological characteristics of the studies, particularly regarding inter-rater reliability of the SNTs, study focus, and type of analysis? (4) What trends can be observed in terms of publication outcomes in studies that use SNTs and integrate them into EHRs? 

Using the initial review as a basis, studies were included if they met the following criteria: (1) primary quantitative design; (2) use of at least one SNT recognized by ANA; (3) reporting on patient and/or organizational outcomes; (4) description of the integration of structured nursing assessment data into EHRs; and (5) sufficient methodological information for analysis.

Question: Are there any patterns in the papers that did not use any SNT
ontologies? Is the initial set of 53 papers sufficient to assess this?

Question: In Table 2, statistical tests are listed and inter-rater reliability counted. But to what are these being applied? Direct tests of the SNT variables or comparisons across patient groups (these are not listed or considered). It is not clear to what the statistics and power analyses are being applied.

Question: Are the SNT variables defined in each study actually used in the statistical models listed in the table? 

Question: Apart from the underlying quality of the study in question and analysis, are the statistical models and tests applied really necessary when examining whether the SNT ontologies are being used to define terms and guide useful data collection? The SNTs may not be part of those analyses.

Question: Gordon’s Eleven Functional Health Patterns (FHPs) were the most frequently used tool (n = 10). Are these 10 among the 17 EHR studies? If not, how many are in the EHR cohort?

Line 172: The interrater reliabilities, are these being carried out at the level of a single SNT with multiple raters, or comparing different SNT descriptions and seeing if they agree? If it is with multiple raters, are the individuals doing the rating all nurses or individuals with comparable training in all studies? The values going from 47% - 98% is a wide range. 

Line 182: Of the 25 studies mentioned it seems that at most 17 contain EHR data. The other 8 studies, are they comparable to the 17 EHR studies?

Do the 10 using the Gordon system overlap with the 17 EHR studies?

The need for standardized data collection and defined terminology is a serious underlying issue for EHR related healthcare research. The paper shows that, from the nursing perspective, there remains a wide variety of symptom descriptions that are not within a formal ontological SNT system.

I think the paper needs to focus on the SNT issues, how the 12 possible ontologies relate to each other, how they relate to non-nursing based collection of medical information.  This should be discussed in more detail. For example do the doctors and clinicians have any exposure to these terms? How can you formally translate data reported by clinicians into nursing oriented terminology? How can you formally translate data collected in one SNT to another SNT? Can you assume that a nurse will be entering all the data?

The lack of a direct focus on EHR in the paper and the inclusion of statistical information that is not particularly useful limits the basic quantitative result here to: 10/53 use the Gordon system and 17/53 used EHR.

There remain issues here in regard to the structure of the paper, the focus of the reported data and the topics being discussed.  Also, as currently written I think the title is misleading.

Author Response

The file containing the response to Reviewer Comment is attached. 

Round 2

Reviewer 1 Report

Comments and Suggestions for Authors

The authors have already revised all of the reviewer's issues. Thank you for carefully revising point-by-point and improving your research. I have no objection. The manuscript is suitable for publication in the Healthcare.

Reviewer 2 Report

Comments and Suggestions for Authors

The paper has been substantially improved.